# Nanoscale imaging of DNA-RNA identifies transcriptional plasticity at heterochromatin

Christelle Guillermier[1], Naveen VG Kumar[2], Ronan C Bracken[3] , Diana Alvarez[4], John O'Keefe[1], Aditi Gurkar[2,4], Jonathan D Brown[5], Matthew L Steinhauser[1,2,6]

**The three-dimensional structure of DNA is a biophysical determinant of transcription. The density of chromatin condensation is one determinant of transcriptional output. Chromatin condensation is generally viewed as enforcing transcriptional suppression, and therefore, transcriptional output should be inversely proportional to DNA compaction. We coupled stable isotope tracers with multi-isotope imaging mass spectrometry to quantify and image nanovolumetric relationships between DNA density and newly made RNA within individual nuclei. Proliferative cell lines and cycling cells in the murine small intestine unexpectedly demonstrated no consistent relationship between DNA density and newly made RNA, even though localized examples of this phenomenon were detected at nuclear–cytoplasmic transitions. In contrast, non-dividing hepatocytes demonstrated global reduction in newly made RNA and an inverse relationship between DNA density and transcription, driven by DNA condensates at the nuclear periphery devoid of newly made RNA. Collectively, these data support an evolving model of transcriptional plasticity that extends at least to a subset of chromatin at the extreme of condensation as expected of heterochromatin.**

## Introduction

A core function of the nucleus is to store the genomic code contained in DNA while also ensuring its accessibility for transcription. Because of the sheer length of the genetic code, DNA is condensed to various degrees to facilitate its storage in the nucleus. This imposes theoretical biophysical constraints on the reading of the genetic code beyond simple 2D interactions between the transcriptional machinery and the DNA sequence. Indeed, remodeling of chromatin architecture resulting in its closure or frank condensation in the form of heterochromatin may silence transcription, demonstrated by both microscopy and genome-scale sequencing methods (Becker et al, 2016; McCarthy et al, 2023; Ochiai et al, 2023).

The generalizability of the chromatin accessibility model as a determinant of transcriptional activity is complicated by numerous studies suggesting active transcription in heterochromatin. In wheat nuclei, foci of transcriptional activity are observable scattered homogeneously throughout the nucleoplasm despite well-demarcated heterochromatic regions (Abranches et al, 1998). So-called "pioneer transcription factors" that bind to DNA within heterochromatic regions have been demonstrated in numerous contexts with chromatin immunoprecipitation (ChIP) among other methods (Zaret & Carroll, 2011; Soufi et al, 2015). ChIP sequencing of bulk cell preparations has demonstrated heterochromatic epigenetic marks and heterochromatic regulatory proteins (e.g., heterochromatin protein-1γ) in association with actively transcribed genes (Vakoc et al, 2005). Transcription of non-coding RNAs in constitutive heterochromatin also appears to play a role in maintenance of heterochromatin structure and inhibition of transcription at pericentromeric heterochromatin gene loci, suggesting that heterochromatic transcription paradoxically enforces transcriptional silencing (Volpe et al, 2002). The complexity of DNA structure–function relationships provides ongoing motivation to expand the armamentarium of methodologies to interrogate chromatin structure and/or function (Cosma & Neguembor, 2023).

Multi-isotope imaging mass spectrometry (MIMS) merges stable isotope tracers with nanoscale secondary ion mass spectrometry (NanoSIMS) (Steinhauser et al, 2012; Steinhauser & Lechene, 2013). MIMS has been used to define subcellular biological processes, including nitrogen fixation within individual bacteria, chromosome segregation patterns during mitosis, dopamine distribution within individual vesicles, protein turnover in hair-cell stereocilia, nuclear pore turnover, and heterogeneity of organellar age in individual cells, among other applications (Lechene et al, 2007; Steinhauser et al, 2012; Zhang et al, 2012; Lovrić et al, 2017; Arrojo e Drigo et al, 2019; Toyama et al, 2019; Narendra et al, 2020). NanoSIMS uses a

[1]Center for NanoImaging, Division of Genetics, Brigham and Women's Hospital and Harvard Medical School, Boston, MA, USA    [2]Aging Institute, University of Pittsburgh School of Medicine, Pittsburgh, PA, USA    [3]Department of Biochemistry, Vanderbilt University School of Medicine, Nashville, TN, USA    [4]Division of Geriatric Medicine, University of Pittsburgh School of Medicine, Pittsburgh, PA, USA    [5]Cardiovascular Division, Vanderbilt University Medical Center, Nashville, TN, USA    [6]Cardiovascular Division, University of Pittsburgh Medical Center, Pittsburgh, PA, USA

Correspondence: msteinhauser@pitt.edu

focused cesium ion beam to sputter atoms and molecular fragments from nanovolumes of a sample surface. The ionized fraction of the ejecta is analyzed with a magnetic sector mass spectrometer, which simultaneously detects several ions from the same nanovolume (Gyngard & Steinhauser, 2019). Incorporation of stable isotope tracers is captured by an increase in the relevant isotope ratio above the natural background. The current NanoSIMS instrument measures up to seven ionic species simultaneously, therefore enabling parallel measurement of up to three isotope ratios. NanoSIMS operates at an optimal lateral resolution down to ~30 nm, which is comparable to super-resolution microscopy (Slodzian et al, 1992). In contrast to high-resolution light microscopy, however, the axial resolution exceeds lateral resolution by at least one order of magnitude. The capacity to quantify multiple tracers in parallel at suborganelle resolution provides a theoretical platform to study structure–function relationships between DNA and transcription within individual nuclei.

The rationale for the current study was to develop a MIMS approach to quantitatively map the relationship between DNA density and transcription. Through multiplexed measurement of DNA labeling intensity together with parallel mapping of $^{15}N$-uridine as a tracer for newly made RNA, we discovered that an inverse relationship between DNA density and RNA synthesis held in limited contexts: (1) in DNA-poor nucleoli where newly made RNA accumulated at a concentration that was up to one order of magnitude higher than the rest of the nucleus, (2) in transcriptionally silent dense DNA at the inner nuclear membrane, and (3) in nonproliferative murine hepatocytes where dense DNA condensates formed that were devoid of newly made RNA. Instead, newly made RNA was detected across the spectrum of DNA density within individual nuclei, particularly in an array of different proliferating cell types. These data therefore provide quantitative support for a model whereby at least a subset of heterochromatin is permissive to transcription and the underlying molecular transcriptional regulatory machinery.

## Results

We previously leveraged MIMS to quantify incorporation of stable isotopically tagged thymidine into the DNA of proliferating cells to define cell turnover dynamics (Steinhauser et al, 2012; Kim et al, 2014). In this study, we sought to quantify RNA synthesis dynamics relative to DNA architecture within individual nuclei by leveraging the multiplexed quantification of $^{15}N$-uridine (RNA) labeling in relationship to DNA, labeled with thymidine tagged with a second isotopic label (either $^{2}H$ or $^{13}C$) (Figs 1A and B, S1A and B, and S2). First, we introduced $^{13}C$- or $^{2}H$-thymidine over two successive passages to proliferating THP-1 cells, a human leukemia monocytic cell line, thereby saturating DNA labeling. Hence, an increase in $^{13}C$ or $^{2}H$ signal within any given intranuclear domain is indicative of a higher DNA concentration relative to regions with a lower $^{13}C$ signal. As in prior studies, single mass images revealed complementary cellular and organellar details (Steinhauser et al, 2012; Guillermier et al, 2017a). $CN^{-}$ or $S^{-}$ images demonstrated cellular and nuclear contours in whole-mount THP-1 cells (Figs 1A and S1). $P^{-}$ images

demonstrated phosphorus-rich chromatin, colocalizing to $^{32}S$-poor regions of the nucleus and correlating with thymidine labeling as previously shown, and which are qualitatively the closest to what is demonstrated by DAPI fluorescent stains (Figs 1B and S1B) (Steinhauser et al, 2012; Guillermier et al, 2017a). We subjected $^{2}H$-thymidine–labeled THP-1 cells to $^{15}N$-uridine pulse in the presence of actinomycin D, a potent inhibitor of RNA synthesis, or vehicle control (Fig 1B). Actinomycin D dramatically neutralized $^{15}N$-uridine labeling, demonstrating the specificity of the $^{15}N$-uridine labeling strategy to capture RNA synthesis. We then performed pulse-chase labeling with $^{15}N$-uridine and detected progressive nuclear labeling between 15 min and 120 min (Fig 1C and D). $^{15}N$-uridine labeling was also evident in the cytosol, though to a lesser degree (Fig 1E). Whereas cytosolic RNA labeling declined with label-free chase, the total mean nuclear RNA labeling remained relatively stable during 120 min of label-free chase. These analyses demonstrate the power of MIMS to quantitatively capture dynamic RNA synthesis and trafficking within individual nuclei.

A notable feature of newly made RNA labeling pattern was its intranuclear heterogeneity with intense $^{15}N$-uridine labeling in structures ranging from nucleoli at ~1–2 $\mu m$ down to small hotspots with lateral dimensions of less than 100 nm. To further define this heterogeneity, we extracted labeling data for each intranuclear pixel from cells labeled for 120 min with $^{15}N$-uridine (Fig 1F). We observed a distribution of isotopic ratios that spanned approximately two orders of magnitude when expressed as percent enrichment above the natural background ratio. The median absolute deviation (MAD) is a statistical metric of dispersion that includes all data points, is not skewed by outliers, and has been previously used to compare relative degrees of heterogeneity in distributions with similar medians (Hampel, 1974; Zhang et al, 2020). During 120 min of label-free chase, intranuclear heterogeneity persisted suggesting organizational forces that may reflect a combination of newly made RNA synthesis and subsequent intranuclear trafficking (Fig 1F).

To test whether MIMS would capture dynamic transcriptional surges, we stimulated THP-1 cells with lipopolysaccharide (LPS) for 120 min concurrently with $^{15}N$-uridine. We also tested the modifying effects of BET bromodomain inhibition through coadministration of JQ1, which targets enhancer-dependent transcriptional responses to inflammatory stimuli and which broadly attenuates new transcripts as measured in bulk samples with SLAM-seq (Brown et al, 2014; Muhar et al, 2018). Indeed, LPS stimulation augmented global transcription by ~50%, an effect partially attenuated by BET bromodomain inhibition (Fig 1G). Collectively, these data demonstrated the power of MIMS to quantify the dynamics of RNA synthesis within individual nuclei, whereas the dramatic intranuclear heterogeneity of newly made RNA labeling is consistent with underlying organizational principles.

We next examined intranuclear organizational features driving heterogeneity of newly made RNA synthesis. At one extreme, we observed spheroidal hotspots of intense $^{15}N$-uridine labeling, consistent with nucleolar patterns first observed decades ago with autoradiographic studies (Fig 2A) (Karasaki, 1965). Putative nucleoli were ~1–3 microns in diameter and DNA-poor at their cores (Fig 2B). The degree of $^{15}N$-uridine labeling was approximately fourfold higher than mean nuclear labeling (Fig 2C), a relationship that increased to approximately sixfold during label-free chase.

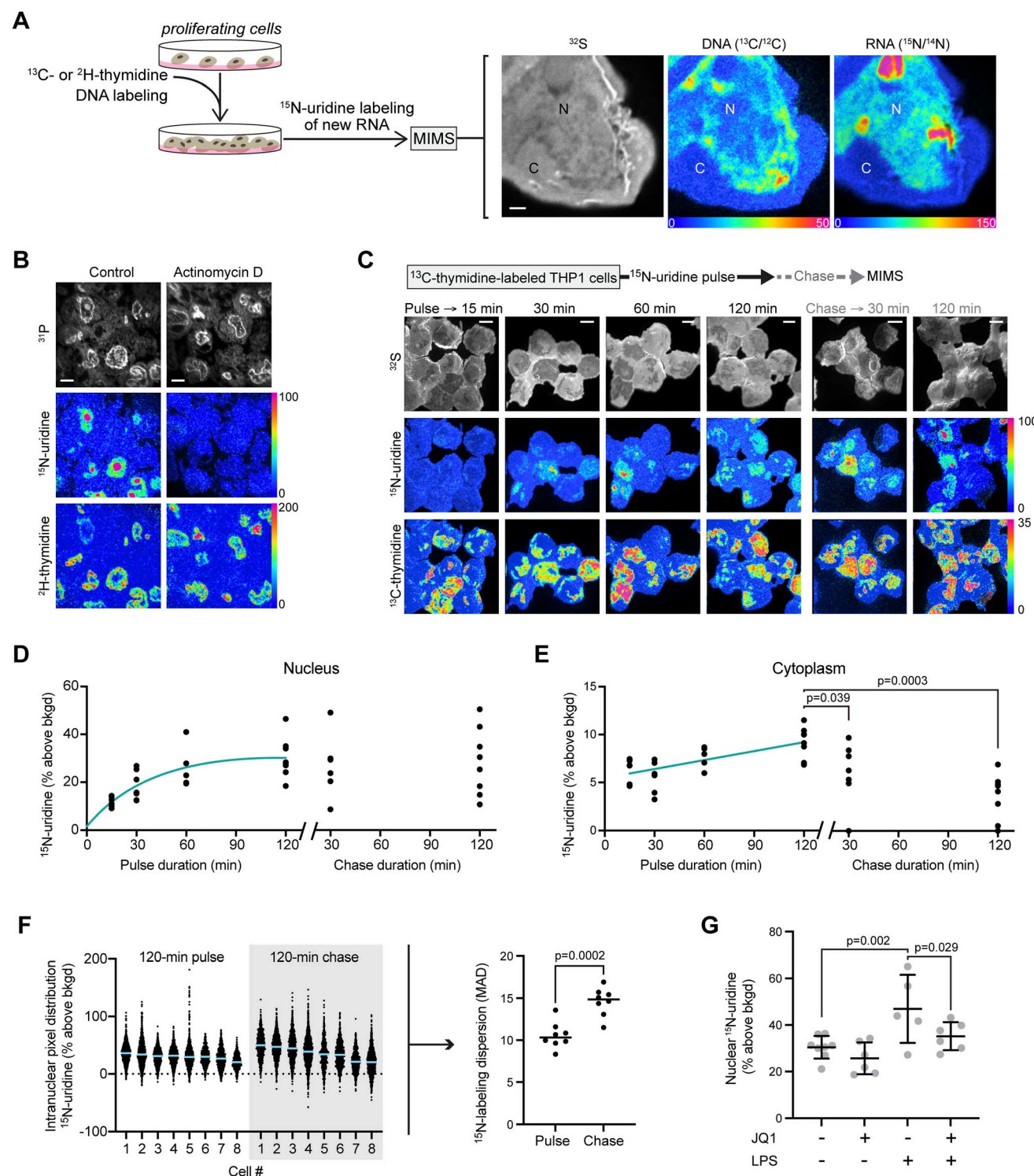

**Figure 1. Multi-isotope imaging mass spectrometry demonstrates intranuclear heterogeneity of new RNA synthesis.**
**(A)** Strategy for multiplexed measurement of new RNA and DNA architecture. $^{13}$C- or $^2$H-thymidine was administered to proliferating cells during serial passage to saturate DNA labeling, followed by pulse labeling with $^{15}$N-uridine as a tracer for new RNA. Cells were analyzed by multi-isotope imaging mass spectrometry with a representative example shown of a THP-1 cell administered with $^{15}$N-uridine for 60 min. $^{32}$S$^-$ images show histological details, including cellular contours, the nucleus (N), and the cytoplasm (C). Hue, saturation, and intensity images display isotope ratio measurements, quantitatively mapping DNA labeling ($^{13}$C/$^{12}$C) and new RNA labeling ($^{15}$N/$^{14}$N). The lower bound of the scale is set to the background ratio (0%), and the upper bound is set to reveal differences in labeling (50% and 150% above background, respectively). Scale bar: 1 $\mu$m. **(B)** $^2$H-thymidine–labeled THP-1 cells pulsed with $^{15}$N-uridine for 120 min with/without RNA synthesis inhibitor (actinomycin D). $^{31}$P mass images (top) show nuclei in a pattern qualitatively similar to DAPI because of the high phosphorous content of chromatin. Scale bar: 5 $\mu$m. **(C)** $^{15}$N-uridine pulse-chase labeling of THP-1 cells. Scale bar: 5 $\mu$m. **(D)** Nuclear quantification of $^{15}$N-uridine pulse-chase labeling. **(E)** Cytoplasmic quantification of $^{15}$N-uridine pulse-chase labeling:

The nuclear lamina at the margin of the nucleus is an intranuclear region where DNA condenses to restrict transcriptional activity. We reasoned that such a phenomenon would manifest as a lag in $^{15}$N-uridine labeling relative to $^{13}$C-thymidine labeling at the cytosolic-to-nuclear margin. Indeed, such a pattern was evident to varying degrees along nuclear margins (Fig 2D). One such nuclear protrusion captured *en face* provided an opportunity to capture a cytosolic-to-nuclear and nuclear-to-cytosolic transition spanning less than three microns and revealing a zone of DNA labeling with little $^{15}$N-uridine labeling, consistent with transcriptional silencing (Fig 2E). To test for this effect at a larger scale, we also performed a comparative analysis in which the nuclear margins were independently traced using $^{13}$C-thymidine labeling (DNA) or $^{15}$N-uridine labeling (RNA) as guidance, finding consistency of the laminar silencing effect (Fig S3). Together, these analyses reveal two contrasting ends of the transcriptional spectrum with a concentration of newly made RNA in the DNA-poor nucleolus and transcriptional silencing at the nuclear membrane contributing to transcriptional heterogeneity. These data further establish the capacity of MIMS to elucidate organizational relationships between the DNA architecture and newly made RNA.

Having demonstrated intranuclear heterogeneity of newly made RNA labeling and predicted regions of transcriptional silencing at the nuclear membrane, we next investigated whether an inverse relationship between DNA condensation and newly made RNA labeling was a more broadly prevalent organizational principle in non-nucleolar regions of the nuclear interior. We first approached this question by identifying labeling hotspots—$^{13}$C-enriched condensed DNA and $^{15}$N-enriched newly made RNA—and tested whether these labeling hotspots were depleted in the complementary label. To test this in a quantitative and unbiased fashion, we leveraged a feature in OpenMIMS software to identify the most intense hotspots, excluding putative nucleoli (Figs 3A and B and S4). As expected, the newly made RNA signal in $^{13}$C-DNA hotspots was generally lower than that observed for the $^{15}$N-RNA hotspots (Fig 3B). Surprisingly, however, the distributions overlapped and the mean $^{15}$N signal in $^{13}$C hotspots was equivalent to or higher than the overall mean of the nucleus. A similar pattern was observed with the complementary analysis of $^{15}$N hotspots, with newly made RNA labeling colocalizing with $^{13}$C-rich regions (Fig 3C). As such, the inverse relationship between $^{15}$N and $^{13}$C labeling that was evident at the nuclear membrane and nucleolus was not consistent throughout the nucleus.

We have previously established that pixel-level isotope ratio data can be used to test for functional associations, as each pixel is a representation of underlying nanovolumetric measurements (Guillermier et al, 2017a; Wertheim et al, 2023). Therefore, we next tested for an inverse relationship between intranuclear DNA condensation ($^{13}$C labeling) and newly made RNA ($^{15}$N labeling), excluding nucleoli (Fig 3D). Using this alternative analysis, we did

not find a consistent correlation between the two labels outside of the nucleolus (slope = −0.93; $R^2$ = 0.2; $P$ < 0.0001) as the slopes of the linear regression models ranged from slightly negative to positive (slopes = −0.005–0.126) with a poor fit ($R^2$=<0.01–0.07) (Fig 3E). We examined whether this relationship held in cells exposed to the transcriptional stimulus of endotoxin (LPS), again finding no consistent correlation between $^{13}$C labeling and $^{15}$N labeling (slopes = −0.08–0.24; $R^2$=<0.01–0.03), although the frequency of pixels exhibiting clearly augmented transcription above control pixels diminished in frequency at the two extremes of DNA density (Fig 3E). These data suggest heterogeneity in the relationship between DNA density and newly synthesized RNA in part driven by evidence of putative newly synthesized RNA in nanovolumes with the highest density of DNA (Fig 3E).

One important consideration for $^{15}$N-uridine labeling of RNA is that it is agnostic to RNA type, inclusive of mRNA and various non-coding RNAs. RNA polymerase III synthesizes short untranslated RNAs, and RNA polymerase I is the dominant transcriber of ribosomal RNA, accounting for approximately half of newly made RNA output. We sought to focus our analyses on newly made RNA synthesized by RNA polymerase II, the activity of which is dominated by synthesis of coding mRNA. As such, we performed pulse labeling in cells administered with inhibitors of Pol I (BMH-21) and Pol III (CAS 577784-91-9) and focused on an early timepoint (15 min), which is earlier than that used in typical measurements of newly made RNA in bulk cell populations with SLAM-seq (Herzog et al, 2017). Inhibition of Pol I/III resulted in a global reduction in $^{15}$N-uridine labeling in line with the known ~50% fractional output of non-coding RNA to total RNA synthesis (Fig 3F). Nucleolar $^{15}$N-uridine labeling dropped dramatically, consistent with the role of RNA Pol I in nucleolar ribosomal RNA synthesis (Fig 3F). We next mapped early hotspots of global newly made RNA (control) and putative newly made mRNA (Pol I/III inhibition) and extracted the DNA labeling intensity. As with longer duration of $^{15}$N-uridine pulse labeling (Fig 3C), we observed hotspots of newly made RNA labeling across the spectrum of DNA density including some that overlapped with DNA intense regions (Fig 3G). Moreover, this pattern was observed in both control cells and cells administered with Pol I/III inhibition, arguing against the restriction of this phenomenon to non-coding RNA.

THP-1 cells are a malignant line, and the patterns observed could reflect transcriptional dysregulation because of cellular transformation. Therefore, we next tested for an inverse correlation between DNA organization and newly made RNA labeling in a non-malignant cell type, human aortic endothelial cells (HAEC). We performed $^{15}$N-uridine labeling of HAEC that had been labeled with $^{13}$C-thymidine during in vitro proliferation. HAEC also demonstrated a heterogeneous $^{15}$N-uridine labeling pattern with similar intense labeling of putative nucleoli (Fig 3H). We again found no consistent association between the degree of DNA condensation and newly

---

pulse compared with chase timepoints by ANOVA with Dunnett's adjusted *P*-values shown. **(F)** Isotope ratio data for intranuclear pixels from 120-min pulse and 120-min chase cells. Left: pixel distribution dot plots with mean line. Right: statistical metric of dispersion (median absolute deviation, MAD) for the pixel distributions with *P*-value reported for an unpaired *t* test. **(G)** Mean nuclear $^{15}$N-uridine labeling (120 min) in THP-1 cells stimulated with lipopolysaccharide and/or the bromodomain inhibitor (JQ1), with significance assessed by two-way ANOVA for overall lipopolysaccharide effect (*P* = 0.0009) and JQ1 effect (*P* = 0.023). Individual *P*-values for multiple comparisons are provided in the figure.

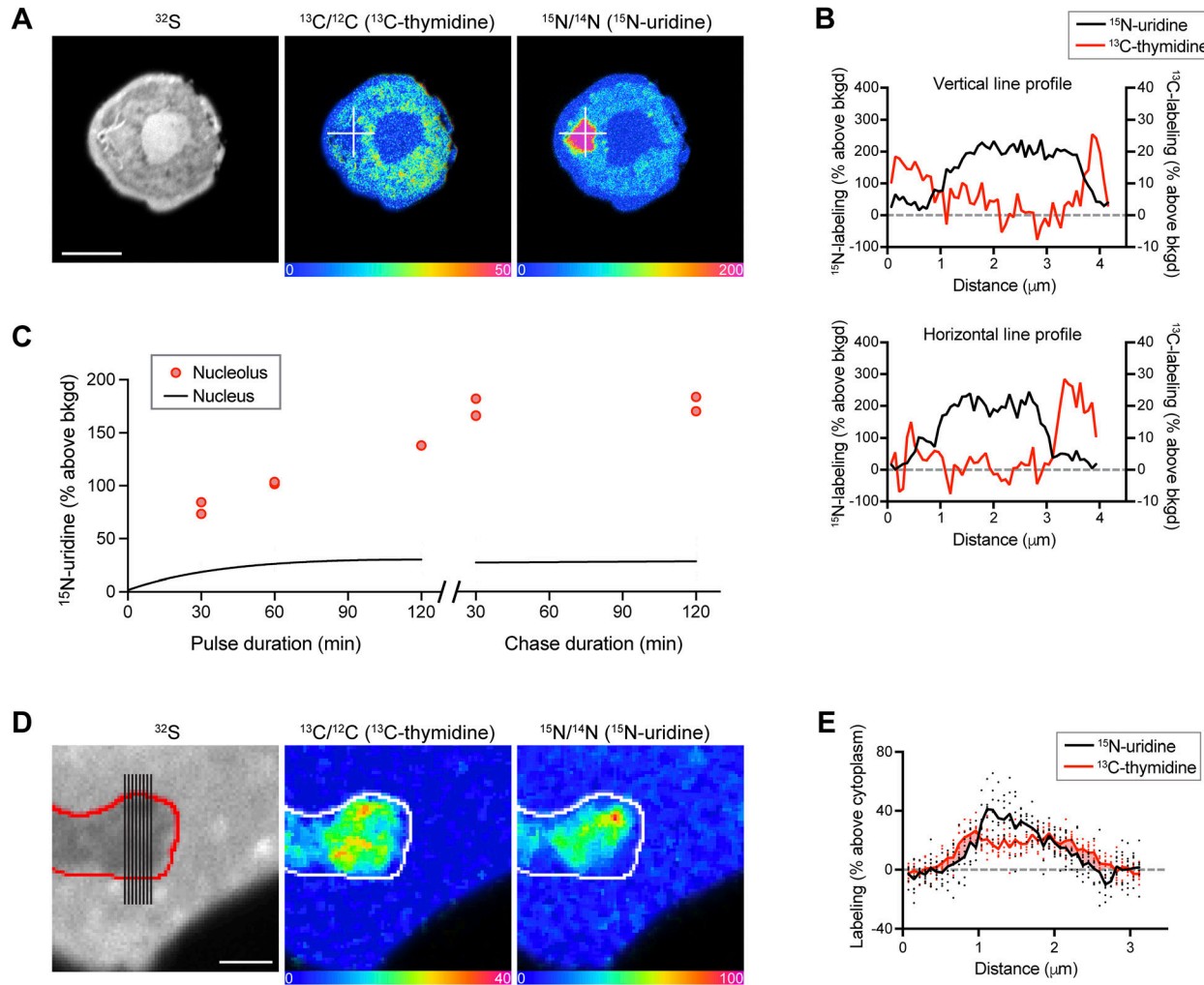

**Figure 2. Multiplexed mapping of DNA density and new RNA reveals intranuclear transcriptional organization.**
**(A)** THP-1 cell after 120 min of $^{15}$N-uridine labeling. The nucleus has a ring shape in the imaging plane. The intense $^{15}$N-uridine signal is seen with the stereotypical size and appearance of a nucleolus. Horizontal and vertical lines shown in the images were used to generate line profiles shown quantitatively in (B). Scale bar: 5 $\mu$m. **(B)** Line profiles for $^{15}$N-uridine (black) and $^{13}$C-thymidine (red) labeling and demonstrating inverse relationship. **(C)** $^{15}$N-uridine pulse-chase labeling of THP-1 nucleoli (dots) relative to the mean signal for the nucleus (black line). **(D)** Nuclear protrusion was captured *en face* in THP-1 cells. Lines shown in the $^{32}$S image (far left) were used to generate line data shown in (E). Scale bar: 2 $\mu$m. **(E)** Lines shown in $^{32}$S image in (D) were used to generate parallel data points from the cytoplasm through the nuclear protrusion and back to the cytoplasm, capturing two transitions across the nuclear membrane and quantitatively showing the lag in new RNA labeling in the DNA at the nuclear membrane. See Fig S3 for additional analyses of labeling at the nuclear lamina.

made RNA with two different durations of $^{15}$N-uridine pulse labeling: 60 and 30 min (Fig 3I). The slopes were generally flat and not directionally consistent at either the 30-min (slopes = −0.05–0.03; $R^2$=<0.01–0.02) or 60-min (slopes = −0.10–0.15; $R^2$=<0.01–0.07) timepoints.

A potential confounding issue with the MIMS analysis is the likelihood that some analyzed intranuclear nanovolumes may contain chromatin in different states, for example, if a nanovolume included material at a hetero-euchromatin transition. We reasoned that pixels (representing nanovolumetric measurements) that contained the highest density $^{13}$C-thymidine signal would be the closest to heterochromatin, and therefore, we performed an analysis of early-timepoint $^{15}$N-uridine pulse labeling and constrained our analysis to the extreme subset of $^{13}$C-labeled pixels (top five percentile) (Fig 3J). In this subset of analyzed nanovolumes

containing the highest DNA density, we still observed newly synthesized RNA ($^{15}$N labeling) detectable over 5–15 min of labeling. These data suggest that the lack of a consistent inverse relationship between the degree of DNA compaction and RNA synthesis was in part due to newly synthesized RNA colocalizing with the regions of the densest DNA.

We next performed high-resolution mapping of newly made RNA, in vivo, by labeling C57BL/6 mice and analyzing resin-embedded and sectioned tissues with MIMS. We used bromodeoxyuridine (BrdU) as a DNA label. Mice were then administered a single dose of $^{15}$N-uridine by intraperitoneal injection and euthanized after 30, 60 min, or 4 h (Fig 4A). We first analyzed liver sections, reasoning that hepatocytes are metabolically active cells with a stereotypical appearance and prominent nuclei. We performed correlative TEM

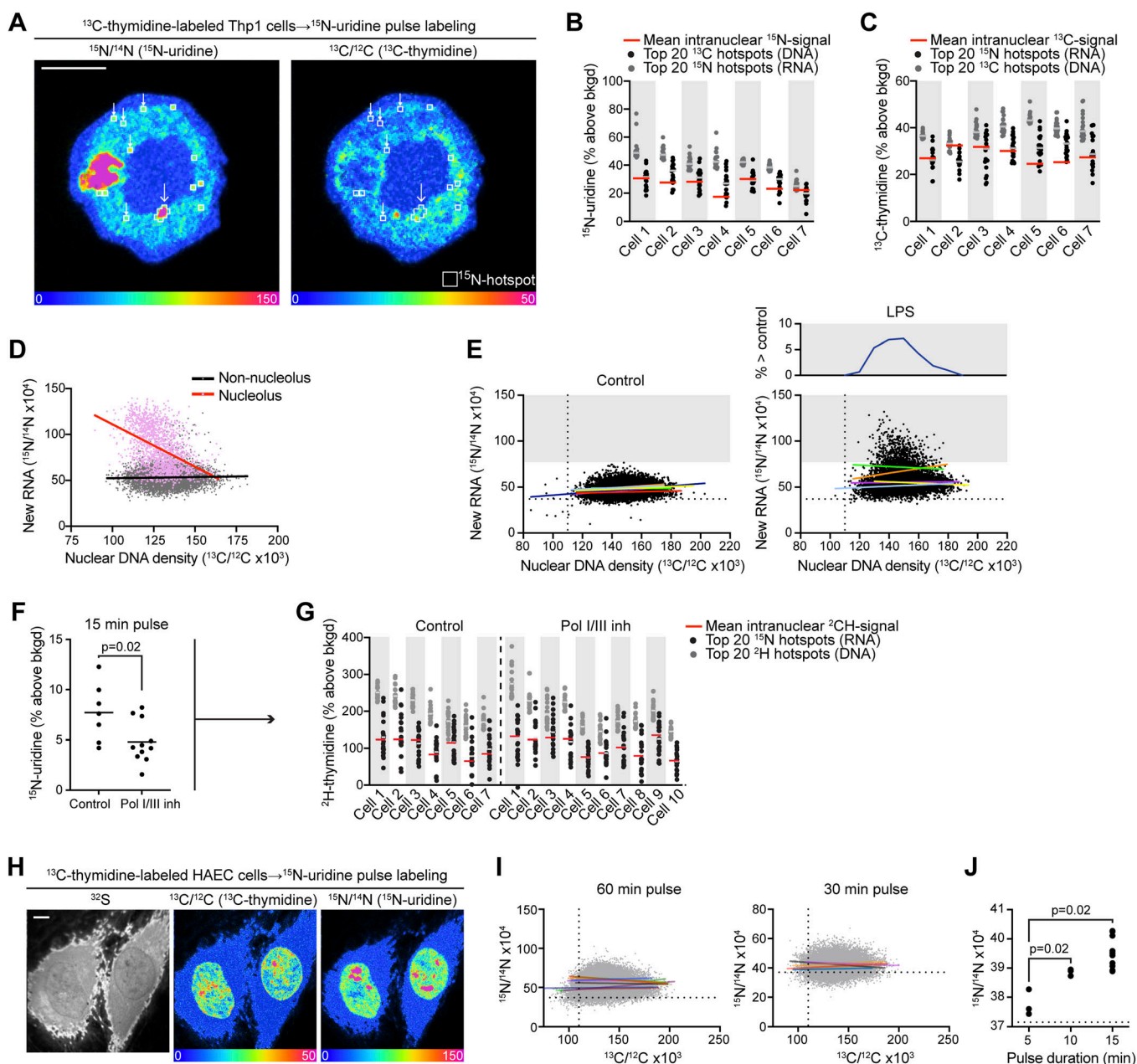

**Figure 3. Detection of new RNA synthesis across the spectrum of DNA density.**
**(A)** Hotspots of new RNA labeling were generated using an automated function in OpenMIMS software (top 20, 4 × 4 pixels). A subset of [15]N-uridine hotpots localized to regions of low DNA labeling (small arrows), whereas others localized to regions of high DNA labeling (large arrow). This cell is the same as that shown in 2A but imaged at greater depth demonstrating the appearance/disappearance of these small puncta in the vertical axis. **(B, C)** Automated hotspot generation was applied to additional cells in (B, C). Scale bar: 5 $\mu$m. **(B)** [15]N labeling distributions for [13]C-DNA hotspots and [15]N-RNA labeling hotspots (n = top 20). A subset of DNA hotspots demonstrates new RNA labeling above the mean of the nucleus (red line) and overlapping with the most intense [15]N-RNA hotspots. **(B, C)** Complementary analysis to (B) showing [13]C labeling distributions for [13]C-DNA hotspots and [15]N-RNA hotspots (n = top 20). A subset of RNA hotspots exhibited a DNA concentration that is above the mean of the nucleus (red line) and overlapping with the most intense DNA hotspots. **(A, D)** Pixel-level correlations for the images shown in (A) demonstrating an inverse correlation between RNA and DNA labeling intensity in the pixels contained in the nucleolus (slope = −0.92; R2 = 0.20; $P < 0.0001$) and a slightly positive but nearly flat correlation for pixels in the non-nucleolar regions of the nucleus (slope = 0.03; R2 = 0.001; $P < 0.0004$). **(A, E)** Non-nucleolar pixel correlations were generated for THP-1 cells labeled with [15]N-uridine for 120 min as in (A). The pixels are merged, but each cell's linear correlation is shown with a different rainbow color. Left: control cells. Right: lipopolysaccharide-stimulated cells. The gray block identifies the range above the highest labeled pixel in the control cells and shows recruitment of new RNA with lipopolysaccharide stimulus across the spectrum of DNA concentration. Like unstimulated cells, there was no consistent correlation between the DNA and RNA signals. **(F)** [2]H-thymidine–labeled THP-1 cells were pulse-labeled with [15]N-uridine for 15 min. To isolate RNA Pol II activity, cells were administered with inhibitors of RNA polymerase I/III (BMH-21 + CAS 577784-91-9 at 1 $\mu$M), which attenuated new RNA, one-way ANOVA, and Sidak's multiple comparison test. **(G)** Hotspots of new RNA synthesis were mapped with the extraction of corresponding [2]H-thymidine labeling intensity. [2]H labeling distributions for [2]H (DNA) hotspots and [15]N-RNA hotspots (n = top 20). A subset of RNA hotspots exhibited a DNA concentration above the mean of the nucleus (red line) and overlapping with intense DNA hotspots. This was observed in both control cells and cells administered with Pol I/III inhibitors. **(H)** [13]C-thymidine–labeled human aortic endothelial cells were pulse-labeled with [15]N-uridine; representative cells labeled for 60 min are shown. Scale

on adjacent sections, which provided an opportunity to compare intranuclear features commonly observed with TEM to their appearance in MIMS mass images (Fig 4B). Darkly stained heterochromatin identifiable by TEM was P-rich and S-poor. Prominent nucleoli in TEM images displayed intense CN emission in tissue sections, a feature previously observed with correlative TEM-MIMS imaging of tissue sections (Kim et al, 2014). Like our in vitro analyses, nucleoli displayed intense $^{15}$N-uridine labeling. Although nuclei were not BrdU-labeled because of the low homeostatic turnover rate of hepatocytes, we observed $P^{high}/S^{low}$ domains that were devoid of newly made $^{15}$N-labeled RNA particularly at the nuclear periphery (Fig 4B). Because non-cycling hepatocytes were BrdU-unlabeled, we used the P/S ratio (Fig S1B) as a proxy for DNA density and assessed the relationship with newly made $^{15}$N-labeled RNA. Non-cycling hepatocytes were the first cell type that exhibited a consistent negative correlation between the newly made RNA signal ($^{15}$N-uridine) and nuclear DNA density (P/S ratio) (Fig 4C and D). However, even though the linear regression slopes were consistently negative, the goodness of fit was generally still poor ($R^2$=<0.01–0.03) suggestive of underlying heterogeneity in the relationship between DNA density and new RNA. The negative relationship between DNA density and new RNA was driven by a paucity of new RNA detected in prominent dense DNA condensates at the nuclear periphery (Fig 4E).

We next turned our attention to the small intestine, reasoning that it would allow study of a proliferative stem/progenitor cell population in the crypt, one that undergoes a well-described cell state transition to terminally differentiated, non-proliferative cells upon migration to the villous epithelium. RNA labeling was dramatically higher in the proliferative crypt cells relative to villous epithelial cells (Fig 4F and G). The reduction in new RNA transcription was accompanied by an increase in heterogeneity of DNA labeling consistent with restructuring of the DNA architecture with terminal epithelial differentiation (Fig 4H).

The intense $^{15}$N labeling of proliferative (BrdU-labeled) crypt cells provided an opportunity to test for an inverse association between DNA density and newly made RNA, using BrdU measurements as a proxy for DNA concentration. Irrespective of the duration after $^{15}$N-uridine pulse, however, we did not detect a consistent inverse relationship with linear regression models exhibiting both positive and negative slopes (Fig 4I–L). By the 4-h timepoint, 10/22 linear models were positive (slopes = −0.25–0.59; $R^2$=<0.01–0.48) (Fig 4K). This was attributable to colocalization of $^{15}$N-uridine hotspots with BrdU-intense foci at both the nuclear margin and nuclear interior (Figs 4L and S5). Seeing as this relationship became more prominent with increased time, it could represent trafficking of non-coding regulatory RNA to regions of dense chromatin.

## Discussion

In this study, we used stable isotopic labeling coupled with high-resolution MIMS to define spatial and quantitative relationships between newly made RNA and DNA concentrations within individual nuclei. Although the nuclear contents are confined by a phospholipid bilayer—the nuclear membrane—there are no equivalent barrier structures within the nucleus. In this context, a striking and consistent finding across cell types was the degree of intranuclear heterogeneity of newly made RNA.

An important principle of the genomics era, elucidated through correlative genomics studies of genome-wide transcription and chromatin states, is that closed and compressed regions of DNA repress transcription, whereas chromatin opening facilitates transcription (Becker et al, 2016). Therefore, our a priori hypothesis was that MIMS would demonstrate an inverse relationship between the nanovolumetric intensity of DNA labeling—a proxy for DNA density—and newly made RNA. Indeed, our examination of multiple different cell types in vitro and in vivo elucidated localized evidence for such an inverse relationship, specifically: (i) intense newly made RNA concentration in DNA-poor nucleoli; (ii) regions of low or minimal newly made RNA in close approximation to the nuclear membrane; and (iii) intensely concentrated DNA hotspots at the nuclear periphery in non-dividing murine hepatocytes that were largely devoid of newly made RNA. Aside from these specific contexts, newly made RNA was generally detectable even in a subset of nanodomains exhibiting the densest DNA signal suggestive of colocalization of new RNA in heterochromatin, with the caveat that the MIMS approach does not define heterochromatin with specific epigenetic markers nor differentiate between constitutive and facultative heterochromatin. $^{15}$N-uridine labeling cannot distinguish between specific types of RNA synthesis, such as the non-coding RNAs previously linked to pericentriolar heterochromatin or gene transcripts driven by pioneering transcription factors (Volpe et al, 2002; Soufi et al, 2015), nor can it resolve newly made RNA at the level of specific loci. However, this study provides quantitative support at single nuclear resolution for an evolving transcriptional model in which heterochromatin may be more plastic and permissive to RNA transcription than previously appreciated (Chereji et al, 2019).

This study also establishes a methodological advance and a new approach to study quantitative and spatial relationships between DNA architecture and transcription in vitro and in vivo that we view as complementary to other methods such as super-resolution microscopy and single-cell genomics. Given extensive precedent for safe administration of stable isotope tracers to humans and prior successful human translation of MIMS to study DNA synthesis and cell turnover, human studies of RNA synthesis and trafficking are also feasible (Steinhauser et al, 2012; Steinhauser & Lechene, 2013; Guillermier et al, 2017a; Yester et al, 2021). Beyond the innocuous nature of stable isotope tracers, several additional strengths of MIMS are worth considering in the context of the application described herein. The imaging resolution of the NanoSIMS instrument (down to <50 nm lateral; <5 nm axial) exceeds that of diffraction-limited light

bar: 5 $\mu$m. **(I)** Non-nucleolar pixel correlations were generated for human aortic endothelial cells after 60 or 30 min of pulse labeling with $^{15}$N-uridine. **(J)** $^{15}$N-RNA labeling of the top 5% of DNA-labeled pixels in human aortic endothelial cells after 5, 10, or 15 min of $^{15}$N-uridine pulse labeling. The dashed line is the measured isotope ratio for an unlabeled cell, which is at the natural background ratio of 37 (x10$^{-4}$).

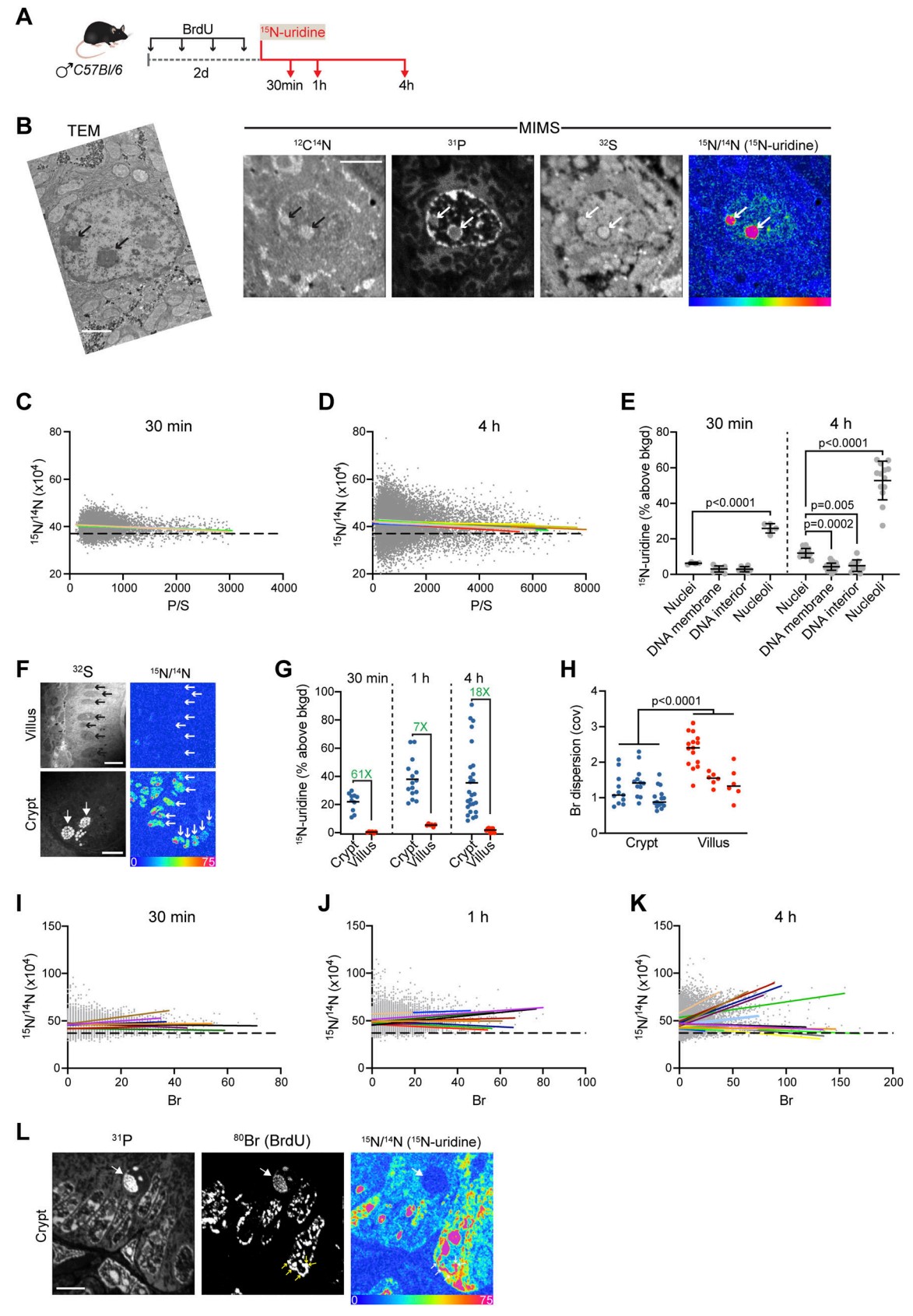

microscopy. The high mass resolution and quantitative power enable the measurement of isotopic ratios with high precision and in turn the incorporation of stable isotope tracers across at least three orders of magnitude (Peteranderl & Lechene, 2004; Steinhauser et al, 2012). Here, we quantitatively imaged two tracers in parallel; however, three labels can be multiplexed (or four with peak switching data acquisition) (Guillermier et al, 2014, 2017b). It is this quantitative power that enabled us to move beyond qualitative colocalization of markers as is typical of fluorescence-based imaging and establish evidence of transcription across the spectrum of DNA density in several different cellular models. In demonstrating how changes in cell state drive nanovolumetric changes in DNA structure–function, this study also provides a template for future applications to study quantitative and spatial relationships between DNA architecture and transcription with mechanistic perturbations or in disease contexts.

# Materials and Methods

## Murine studies

Stable isotope labeling protocols were approved by and in compliance with the University of Pittsburgh Institutional Animal Care and Use Committee. Mice were maintained under a 12-h dark/light cycle at 22°C ± 2°C and administered food and water ad libitum. Stable isotopes were purchased from Cambridge Isotope Laboratories. Bromodeoxyuridine (BrdU) was purchased from Sigma-Aldrich. Mice were euthanized with a lethal dose of $CO_2$ and perfused with 10 ml cold PBS, followed by 10 ml cold 4% PFA. Liver and small intestine samples were embedded in EPON and serially sectioned for TEM and MIMS. TEM samples were ~80 nm thick and mounted on slot grids. MIMS samples were 500 nm thick and mounted on silicon wafers.

## Cell culture

THP-1 cells (ATCC) and human aortic endothelial cells were cultured as previously described (Fazeli et al, 2020). Briefly, THP-1 cells were cultured in suspension culture, using RPMI 1640 (Corning)

supplemented with 10% FBS (Corning). Labeling studies were performed in suspension culture, and then, the cells were washed, fixed (4% PFA), and either smeared onto silicon wafers or embedded (EPON), sectioned, and mounted on silicon wafers (cell pellet sections = 500 nm). hTERT-immortalized human aortic endothelial (TeloHAEC) (CRL-4052; ATCC) cells were cultured in Vascular Cell Basal Medium (PCS-100-030; ATCC) supplemented with Endothelial Cell Growth Kit-VEGF (PCS-100-041; ATCC) containing $^{13}C$-thymidine (50 mM) and $^{15}N$-uridine (50 mM) as previously applied for measurement of global cellular changes in $^{15}N$-uridine labeling (Bracken et al, 2024). BMH-21 and CAS 577784-91-9 were purchased from Calbiochem, Inc. Adherent endothelial cells were seeded onto silicon wafers with the final passage before $^{15}N$-uridine labeling. After $^{15}N$-uridine pulse labeling, the silicon wafers were fixed with 4% PFA, dehydrated in ascending concentrations of ethanol, and then air-dried.

## MIMS

Samples were analyzed with a NanoSIMS 50L instrument (CAMECA), using previously published analytical methods. Although the idealized resolution of the instrument can get down to a lateral resolution of <30 nm, the standard operation resolution for the analyses contained in this study is in the ~40–120 nm range. $^{13}C$-thymidine labeling was measured either by the $^{13}C^{12}C^-/^{12}C_2^-$ ratio or by the $^{13}C^{14}N^-/^{12}C^{14}N^-$ ratio; $^2H$-thymidine labeling measured by the $^{12}C_2^2H^-/^{12}C_2^1H^-$ ratio; and $^{15}N$-uridine labeling by the $^{12}C^{15}N^-/^{12}C^{14}N^-$ ratio as previously described (Kim et al, 2014; Guillermier et al, 2017b). BrdU labeling was measured as previously described by direct capture of $^{81}Br^-$ (Steinhauser et al, 2012; Zhang et al, 2020). The instrument was also tuned to capture $^{31}P^-$ and $^{32}S^-$. Image files were visualized and analyzed with a custom plugin to ImageJ (OpenMIMS 3.0: https://github.com/BWHCNI/OpenMIMS; Guillermier et al, 2017b). $^{32}S^-$ and $^{13}C^-$ or $^2H$ labeling images were used to guide the manual selection of regions of interest (ROIs) corresponding to the nucleus, the cytoplasm (excluding the nucleus), or other stereotypically identifiable structures. For automated hotspot selection, OpenMIMS software identifies square ROIs with defined lateral dimensions ranked by isotope ratio for ion counts contained within the boundaries of the ROI. For all ROIs, whether

**Figure 4. New RNA across the spectrum of DNA density in proliferating cells in vivo.**
**(A)** Schematic of murine labeling protocol (i.p. injection). BrdU was administered for 2 d to allow for labeling of crypt cells and their epithelial progeny before a single $^{15}N$-uridine pulse. **(B)** Liver (hepatocyte) imaged 4 h after $^{15}N$-uridine. Adjacent liver sections were imaged by TEM (left) and multi-isotope imaging mass spectrometry (MIMS) (right). Intense $^{15}N$-uridine–labeled nucleoli (arrows) are evident by TEM and MIMS. Scale bar: TEM = 2 $\mu$m. Scale bar: MIMS = 5 $\mu$m. **(C)** Pixel correlations in 30-min hepatocyte nuclei, excluding nucleolar regions. The P/S ratio was used as a proxy for DNA density. Each linear regression line has a negative slope and is shown in a different color (n = 4 nuclei). A dashed line was set at a natural abundance ratio of 0.0037. **(D)** Pixel correlations in 4-h hepatocyte nuclei, excluding nucleolar regions. The P/S ratio was used as a proxy for DNA density. Each linear regression line has a negative slope and is shown in a different color (n = 11 nuclei). **(E)** Mapping of hepatocyte intranuclear new RNA 30 min and 4 h after $^{15}N$-uridine administration. At both timepoints, P-intense putative DNA at the nuclear membrane and the interior of the nucleus was minimally labeled, whereas the nucleoli were highly labeled. **(F)** $^{15}N$ labeling of new RNA in small intestine 30 min after injection, revealing the dramatic difference in nuclear (small arrows) labeling intensity between proliferative crypt cells and terminally differentiated villus epithelial cells. Large arrows show sulfur-rich granules in Paneth cells at the base of the crypt. Scale bars = 10 $\mu$m. **(G)** Intranuclear new RNA labeling in proliferating crypt cells versus non-proliferating villus epithelial cells, 30 min, 1 h, and 4 h after I.P. injection of $^{15}N$-uridine. The fold difference is shown in green. **(H)** Metric of dispersion (COV) demonstrates increased heterogeneity of the DNA distribution in villus cells relative to crypt cells. Nested t test. **(I)** Pixel correlations in BrdU-labeled crypt nuclei 30 min after $^{15}N$-uridine pulse, excluding nucleolar regions. **(J)** Pixel correlations in BrdU-labeled crypt nuclei 1 h after $^{15}N$-uridine pulse, excluding nucleolar regions. **(K)** Pixel correlations in BrdU-labeled crypt nuclei 4 h after $^{15}N$-uridine pulse, excluding nucleolar regions. **(L)** Proliferating crypt cells 4 h after $^{15}N$-uridine. The large arrow shows BrdU-labeled mitotic figure that is devoid of new RNA. Small arrows show regions of densely BrdU-labeled DNA that are also intensively labeled with $^{15}N$-uridine, accounting for a subset of nuclei demonstrating a positive correlation between DNA density and new RNA in (J, K). Scale bar = 5 $\mu$m.

generated manually or by automated selection, the corresponding isotope ratios were extracted from the pixels contained within each respective ROI. Isotope ratio data are displayed as hue, saturation, and intensity images (Fig S2). The lower bound of the scale (blue) was set at the natural background (e.g., for $^{15}$N-uridine data, a lower bound of 0 is equivalent to the natural background of 0.37% = no labeling, and an upper bound of 100 corresponds to a ratio of 0.74%). For representative images, the upper bound of the scale was set to demonstrate regional differences in labeling. Importantly, the underlying quantitative data are unmodified by changes in the scaling to displayed images. Throughout the figures, when isotope ratio data are expressed as a raw ratio (as opposed to % above background), it is written ×$10^4$ (e.g., background ratio for $^{15}$N/$^{14}$N of 0.0037 is reported as 37 for ease).

## Statistics

Statistical analyses were performed with Prism 10 (GraphPad) with two-sided alpha < 0.05.

# Supplementary Information

# Acknowledgements

The authors thank Louise Trakimas in the Harvard Medical School Electron Microscopy Core for assistance with sample processing and transmission electron microscopy. The work was funded by discretionary funds from the University of Pittsburgh School of Medicine (to ML Steinhauser), NIH DP2CA216362 (to ML Steinhauser), and NIH R01HL146654 (to JD Brown).

## Author Contributions

C Guillermier: formal analysis, investigation, methodology, and writing—review and editing.
NVG Kumar: investigation and writing—review and editing.
RC Bracken: investigation and writing—review and editing.
D Alvarez: investigation and writing—review and editing.
J O'Keefe: investigation and writing—review and editing.
A Gurkar: supervision, investigation, and writing—review and editing.
JD Brown: supervision, funding acquisition, investigation, and writing—review and editing.
ML Steinhauser: conceptualization, formal analysis, supervision, funding acquisition, investigation, visualization, methodology, project administration, and writing—original draft, review, and editing.

## Conflict of Interest Statement

The authors declare that they have no conflict of interest.

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
