## [Reviewer comments · Life Science Alliance]

Nanoscale imaging of DNA-RNA identifies transcriptional plasticity at heterochromatin

Christelle Guillermier, Naveen VG Kumar, Ronan C Bracken, Diana Alvarez, John O'Keefe, Aditi Gurkhar, Jonathan D Brown and Matthew L Steinhauser
DOI: 10.26508/lsa.202402849

Corresponding author(s): Prof. Matthew Steinhauser (University of Pittsburgh)

Review timeline:

Submission Date:	2024-05-29
Editorial Decision:	2024-07-25
Revision Received:	2024-08-30
Editorial Decision:	2024-09-04
Revision Received:	2024-09-05
Accepted:	2024-09-06

Scientific Editor: Eric Sawey

Transaction Report:

No Peer Review Process File is available with this article, as the authors have chosen not to make the review process public in this case.

1st Editorial Decision

25 July 2024

Re: Life Science Alliance manuscript #LSA-2024-02849-T

Matthew Steinhauser

University of Pittsburgh School of Medicine Aging Institute

Dear Dr. Steinhauser,

Thank you for submitting your manuscript entitled "Nanoscale imaging and quantification of DNA-RNA relationships identifies transcriptional plasticity at heterochromatin" to Life Science Alliance. The manuscript was assessed by expert reviewers, whose comments are appended to this letter. We invite you to submit a revised manuscript addressing the Reviewer comments.

Thank you for this interesting contribution to Life Science Alliance. We are looking forward to receiving your revised manuscript.

Sincerely,

Eric Sawey, PhD

Executive Editor

Life Science Alliance

<http://www.lsjournal.org>

B. MANUSCRIPT ORGANIZATION AND FORMATTING:

RE: Life Science Alliance Manuscript #LSA-2024-02849-TR

Prof. Matthew Steinhauser
University of Pittsburgh
100 Technology Drive
Rm 558
Pittsburgh, PA 15219

Dear Dr. Steinhauser,

Thank you for submitting your revised manuscript entitled "Nanoscale imaging of DNA-RNA identifies transcriptional plasticity at heterochromatin". We would be happy to publish your paper in Life Science Alliance pending final revisions necessary to meet our formatting guidelines.

- please be sure that the authorship listing and order is correct
- please upload your supplementary figures as single files
- please add the Twitter handle of your host institute/organization as well as your own or/and one of the authors in our system
- please make sure that the author order in the manuscript text and the author order in our system match
- please add the author contributions to your main manuscript text
- please consult our manuscript preparation guidelines <https://www.life-science-alliance.org/manuscript-prep> and make sure your manuscript sections are in the correct order (please add a separate figure legend section for all figures and note that the results and discussion sections should be separate)
- please add the panel E to your Figure 2 figure legend
- there is a figure callout for Figure 1H, but this panel is not in the figure or the legend-please correct
- please add a figure callout for Figure 2E, Figure 3H, Figure 4L and Figure S1A

Figure Check:

- please add scale bars to Figure 3H; Figure 4B,F; Figure S1A; Figure S2; Figure S3; Figure S4; Figure S5

LSA now encourages authors to provide a 30-60 second video where the study is briefly explained. We will use these videos on social media to promote the published paper and the presenting author (for examples, see <https://docs.google.com/document/d/1-UWCfbE4pGcDdcgzcmiuJI2XMBJnxKYeqRvLLrLS08s/edit?usp=sharing>). Corresponding or first-authors are welcome to submit the video. Please submit only one video per manuscript. The video can be emailed to contact@life-science-alliance.org

A. FINAL FILES:

B. MANUSCRIPT ORGANIZATION AND FORMATTING:

Sincerely,

3rd Editorial Decision

21 October 2019

RE: Life Science Alliance Manuscript #LSA-2024-02849-TRR

Prof. Matthew Steinhauser
University of Pittsburgh
100 Technology Drive
Rm 558
Pittsburgh, PA 15219

Dear Dr. Steinhauser,

Thank you for submitting your Research Article entitled "Nanoscale imaging of DNA-RNA identifies transcriptional plasticity at heterochromatin". It is a pleasure to let you know that your manuscript is now accepted for publication in Life Science Alliance. Congratulations on this interesting work.

DISTRIBUTION OF MATERIALS:

Again, congratulations on a very nice paper. I hope you found the review process to be constructive and are pleased with how the manuscript was handled editorially. We look forward to future exciting submissions from your lab.

Sincerely,
